# Successful Treatments and Management of A Case of Canine Melioidosis

**DOI:** 10.3390/vetsci6040076

**Published:** 2019-09-23

**Authors:** Pacharapong Khrongsee, Chulalak Lueangthuwapranit, Thitsana Ingkasri, Somporn Sretrirutchai, Jedsada Kaewrakmuk, Vannarat Saechan, Apichai Tuanyok

**Affiliations:** 1Faculty of Veterinary Science, Prince of Songkla University, Hatyai, Songkhla 90110, Thailand; firstpachar@ufl.edu (P.K.); Chulalak494@gmail.com (C.L.); Ingkasri@gmail.com (T.I.); Svannarat2002@yahoo.com (V.S.); 2Department of Infectious Diseases and Immunology, College of Veterinary Medicine, and the Emerging Pathogens Institute, University of Florida, Gainesville, FL 32608, USA; 3Faculty of Medicine, Prince of Songkla University, Hatyai, Songkhla 90110, Thailand; Sompornsre@gmail.com; 4Faculty of Medical Technology, Prince of Songkla University, Hatyai, Songkhla 90110, Thailand; jedsada.k@psu.ac.th

**Keywords:** canine melioidosis, *Burkholderia pseudomallei*, meropenem, sulfamethoxazole-trimethoprim

## Abstract

This communication presents a successful story of an attempt to treat and manage a case of canine melioidosis, a severe tropical disease caused by *Burkholderia pseudomallei.* A 10-year-old dog was trapped with barbed wires, causing an infected wound around its neck and back, which was later diagnosed as severe melioidosis. The dog was treated based on a modified human protocol. Intravenous meropenem injections (20 mg/kg twice daily) were given for 14 days to prevent death from sepsis prior to treatment with oral sulfamethoxazole-trimethoprim (25 mg/kg twice daily) for 20 weeks to eliminate the bacteria. Canine melioidosis is an unusual infection in dogs, even in Thailand where melioidosis is highly endemic. This successful case management was solely based on proper diagnosis and appropriate treatments.

## 1. Case Presentation

In August 2018, a 10-year-old male Pomeranian–mixed breed dog from Pattani, a southern province of Thailand, was referred from a local animal clinic to our hospital at Prince of Songkla University (PSU) in Hatyai, Songkhla Province. At admission, the owner reported that seven days earlier, the dog had run away from home at night and probably got hit by barbed wires at a nearby army camp. In the morning, the dog was found with large dirty wounds around its neck and back. The dog was immediately taken to a local animal clinic for primary care. The dog was admitted and then given primary treatment as well as wound dressing daily. The primary treatment included oral amoxicillin–clavulanic acid (30 mg/kg) twice daily and oral carprofen (4 mg/kg) once daily. Four days later, the wounds got worse with progressive purulent discharge and odor. An additional antibiotic, metronidazole (18 mg/kg), was then given twice daily. The blood profile examined on that day showed a high number of white blood cells (22,400 cells/mL) and low platelets (33 × 10^9^ PLTs/L). After seven days of treatments, the wounds continued to expand. The dog was then transferred almost 100 miles to PSU Animal Hospital, a teaching hospital in the region, for proper laboratory diagnosis and appropriate treatment. 

At the second admission, a physical examination and laboratory testing were conducted. The physical examination revealed that the dog was slightly depressed with a high body temperature of 102.8 ^๐^F (despite treatment with the anti-inflammatory drug from the previous hospital), pale mucus membranes in its mouth, and severe seropurulent discharge around its large cervical wounds (Figure 1A). Blood profile analysis revealed leukocytosis (35,940 WBCs/mm^3^), severe anemia (14.1% hematocrit), extremely severe thrombocytopenia (22 × 10^9^ PLTs/L), slightly increased liver enzyme (ALP 192 U/L), and positive serological detections of two common bacterial pathogens, *Ehrlichia canis* and *Anaplasma platys,* by a rapid test SNAP 4Dx Plus (IDEXX Inc., Westbrook, ME, USA). The dog was then intravenously treated with enrofloxacin (10 mg/kg) once daily, aiming to kill the blood parasites as well as providing broad-spectrum antibacterial activity while awaiting the results from culture and biochemical tests. Three days later, a second complete blood count was suggested since clinical signs persisted. The blood profile still showed progressive severe leukocytosis (74,580 WBCs/mm^3^), thrombocytopenia, and increased liver enzymes. The culture results from the wound swabs revealed the presence of *Proteus mirabilis, Pseudomonas aeruginosa,* and *Escherichia coli,* all of which were resistant to both previously used antibiotics, while the hemoculture grew *Burkholderia pseudomallei* (Figure 1B), the causative agent of melioidosis.

## 2. Melioidosis Case Management

Since melioidosis is a notifiable disease in Thailand, a local animal health authority was notified for this case. The dog was then isolated into a clean room, and standard precautions were applied to reduce the risk of transmission of the pathogen from the infected dog to other animals and humans. We also noted that there were no guidelines for treating canine melioidosis in veterinary practice in Thailand or elsewhere. Therefore, the treatments were selected based upon those recommended and available for treating human melioidosis. Intravenous meropenem (20 mg/kg twice daily) was chosen as the primary treatment due to the unavailability of ceftazidime, the most recommended antibiotic for treating acute melioidosis in Thailand. During the treatment, blood profile analysis and hemoculture were repeated every three days. Blood cultures were all negative for *B. pseudomallei* after the initiation of treatment. The indirect hemagglutination (IHA) test was also used to identify the antibody titer for melioidosis. The dog’s antibody titer was determined at 1:640 to be serologically positive for melioidosis. Ultrasonography and X-ray imaging were conducted and identified an abscess-like mass at the splenic tail (Figure 1C). Although the spleen is a common site for disseminated abscess formation in melioidosis, no attempt was made to confirm that the splenic abscess in the dog was indeed due to *B. pseudomallei* infection, even though that seems likely. In addition, the wounds, urine, feces, and fomites were also cultured using Ashdown’s medium to identify the shedding of *B. pseudomallei* from the infected dog, but all showed negative results. The minimal inhibitory concentration (MIC) test of meropenem and sulfamethoxazole–trimethoprim (co-trimoxazole) on the *B. pseudomallei* isolate was performed using the E-Test based on CLSI standards, which confirmed susceptibility to both drugs, with an MIC of 1 µg/mL. The dog gradually recovered from the acute infection after being treated with meropenem for 14 days with no cultures positive from either blood or other clinical specimens. Since the *B. pseudomallei* isolate was susceptible to co-trimoxazole, a recommended oral treatment for melioidosis, the regimen was then changed to eradication phase treatment with oral co-trimoxazole, 25 mg/kg twice daily for 20 weeks. In addition to co-trimoxazole, in the first 3 weeks of the eradication phase, oral doxycycline (10 mg/kg) was given once daily to treat *Ehrlichia canis* and *Anaplasma platys,* since both bacterial pathogens were still found in the blood based on the serological detections. The dog stayed in the hospital for 5 additional weeks until the largest wound had recovered completely. The dog was then discharged to continue the oral treatment (co-trimoxazole) at home. The dog was re-evaluated by blood culture at the first follow-up, 4 weeks after discharge, showing negative results. The blood chemistry also appeared to be normal and the mass-like abscess in the spleen had disappeared at the second follow-up (Figure 1D). The dog remained on the oral treatment with co-trimoxazole for a total of 20 weeks. All human contacts with the dog were also monitored by serological testing for melioidosis using IHA. All the contacts showed seronegative results.

## 3. Environmental Source of the Infection 

A total of 176 soil samples and seven water samples were collected from the owner’s homesite and cultured for *B. pseudomallei* using the consensus guidelines as previously described [1]. There was no evidence of *B. pseudomallei* in these samples. Genomic DNA of the *B. pseudomallei* isolate from the dog was sent to the Emerging Pathogens Institute at University of Florida for genetic characterization. Multi-locus sequence typing (MLST) was conducted as previously described [2] and showed the sequence type (ST) 366. Based on the current MLST database (https://pubmlst.org/bpseudomallei/), ST366 is a local genotype of *B. pseudomallei* that has been found in the environment and human cases in southern Thailand. We noted that this ST has also been found in Vietnam and China. 

## 4. Discussion

*B. pseudomallei*, a soil-dwelling Gram-negative bacterium, causes melioidosis, a deadly, infectious disease in the tropics. Melioidosis is highly endemic in Thailand with approximately 2800 deaths or 35% mortality in treated patients throughout the country annually [3]. Although melioidosis is known to be endemic in most parts of Thailand, information about this disease, especially in southern Thailand where the canine infection described here occurred, is still limited. The disease is rarely reported in animals. The most recent report from Thailand showed incidence rates in goats, pigs, and cattle of approximately 0.02–1.6/100,000/year [4], with no cases in dogs. The lack of reports of melioidosis in dogs probably reflects a lack of testing and surveillance in most companion animals. Melioidosis in dogs, however, may have clinical symptoms similar to those observed in humans. The presence of abscess, pneumonia, or even asymptomatic seroconversion have been reported in dogs [5,6]. A previous case of canine melioidosis was reported in a 5-week pregnant Doberman living in a farm in Australia where melioidosis outbreak was ongoing in goats [5]. That dog developed a bloody discharge from the vulva accompanied by anorexia, dehydration, depression, and fever. The pregnancy was terminated due to the possibility of intra-uterine sepsis causing the clinical signs. After abortion, the dog showed signs of abdominal pain, and numerous cutaneous abscesses were developed. Euthanasia was performed and a skin lesion swab cultured *B. pseudomallei*. 

To the best of our knowledge, euthanasia is recommended in most melioidosis cases in pets. In a recent report, the euthanasia was also recommended as a public health response to a case of subclinical urinary *B. pseudomallei* infection in a dog that was adopted into upstate New York from a shelter in Thailand due to environmental contamination concerns [7]. However, in our case in southern Thailand, treatment was selected, rather than euthanasia. Although, *B. pseudomallei* has a zoonotic potential, transmission of the pathogens from an animal to human is extremely unusual [8].

Hemoculture is uncommon in routine veterinary diagnosis in most animal hospitals in Thailand. Without hemoculture, we may not be able to identify the cause of infection for this case. Thus, the veterinarians may consider using hemoculture to diagnose dogs or other companion animals with febrile illness. Although the guidelines for treating canine melioidosis are not currently available in veterinary practice in Thailand, our report may be used as a guide for canine melioidosis case management with cautions that this report was based on a single case study. Importantly, we have demonstrated that the treatments for both acute and eradication phases can follow the guidelines for human melioidosis [9]. The management of drug administration to overcome this disease depends on both doses and times. Therefore, maintaining the drug concentration above the MIC level is important. It has been shown in dogs that meropenem can be administered at a dose of 8 mg/kg every 12 h subcutaneously in order to achieve adequate tissue fluid and urine concentrations for susceptible bacteria with a minimum inhibitory concentration of 0.12 µg/mL [10]. The recommended treatment concentration of ceftazidime for dogs is 30 mg/kg subcutaneously every 8 h or continuous IV infusion of ceftazidime (loading dose, 4.4 mg/kg; infusion rate, 4.1 mg/kg/h) in patients with no complications [11]. Also, meropenem (25 mg/kg) can be administered subcutaneously three times a day in intensive care canine patients [10]. To the best of our knowledge, both meropenem and ceftazidime are not given subcutaneously in humans. In human melioidosis, it is also suggested that blood cultures should be repeated weekly until they become negative in patients with bacteremic melioidosis, and treatment should be extended if cultures are positive; normally 10–14 days are required [12]. It has also been discussed elsewhere that the use of carbapenem antibiotics (e.g., meropenem, imipenem) may raise some ethical concerns about resistance since it should be restricted to severe cases of human bacterial infection. Using this class of antibiotics to treat melioidosis in animals will still need further studies. 

The eradication of *B. pseudomallei* is also difficult due to the high rate of relapse if therapy is not completed; thus, prolonged antibiotic therapy is essential. In a previous report, short-term administration of tetracycline at high dosage levels appeared to be a treatment of choice for canine melioidosis, but the dog was eventually euthanized [13]. For human cases, it is important to treat patients with antibiotics for up to 20 weeks to prevent relapse. In the past, it was recommended to use a combination of doxycycline and co-trimoxazole. The recent study, however, revealed that the treatment of choice for the eradication phase of melioidosis is co-trimoxazole monotherapy due to the adverse effects of long-term doxycycline use [14]. It is also relevant to monitor the adverse drug effect and liver enzymes during combination therapy.

Genotyping of the bacterial pathogen was also performed as part of the epidemiological investigation. We noted, using MLST, that the dog was infected with a local genotype of *B. pseudomallei* that has been found in the environment in southern Thailand. We could not isolate this pathogen from the environmental sources collected at the owner’s yard. Although the wound swabs grew other opportunistic bacteria, not *B. pseudomallei*, it is likely that the dog contracted the disease through the skin injuries. It is obvious that other opportunistic bacteria grew faster than *B. pseudomallei*, even though a selective agar, Ashdown’s agar, was used. 

In conclusion, successful treatment of this case of canine melioidosis was likely the result of accurate and prompt diagnosis and the application of appropriate treatment. 

## Figures and Tables

**Figure 1 vetsci-06-00076-f001:**
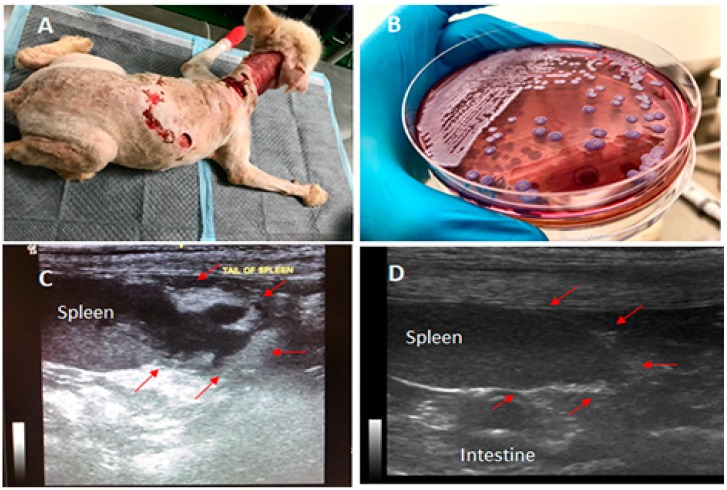
Canine melioidosis: **A**, showing the massive wounds around the dog’s neck and back observed on the 1^st^ day at Prince of Songkla University (PSU) Animal Hospital; **B**, *Burkholderia pseudomallei* colonies grown on Ashdown’s selective medium; **C**, the presence of a 2-cm abscess-like mass at the splenic tail confirmed by ultrasonography; and **D**, the recovery of the splenic tail after 10 weeks of the oral treatment. The arrows point to the areas of abscesses observed in C, which later disappeared after the oral treatment.

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
