# Peer review of "Successful Treatments and Management of A Case of Canine Melioidosis"

_vetsci, 2019, doi:10.3390/vetsci6040076_

Round 1

Reviewer 1 Report

General comment:

This paper describes an interesting case of canine melioidosis successfully treated. The manuscript is overall well written ond should be worthy of publication after a minor revision. My major concern is about the use of an antibiotic (meropenem) that in some countries is forbidden for the use on veterinary medicine. The use of this class of antibiotics raise some ethical problem since it should be restricted to severe cases of human bacterial infection. This matter must be discussed further by the authors.

Specific comments:

Title: I suggest to change as “Successful treatments and management of a case of canine melioidosis”

Line 49: please specify how the detection of E. canis and Anaplasma was done (Serology? PCR? Microscopic detection? If so please provides images). Moreover, I am not aware of Anaplasma canis species, rather Anaplasma phagocytophilum?

Line 94-95: again, specify how these agents were detected. Moreover, these two agents are not parasites but bacteria, please correct in the text

Line 142-142: this sentence claims a general conclusion which is however based on a single case report. I suggest to be more cautious.

Author Response

Reviewer 1:

Comments and Suggestions for Authors

General comment:

This paper describes an interesting case of canine melioidosis successfully treated. The manuscript is overall well written and should be worthy of publication after a minor revision. My major concern is about the use of an antibiotic (meropenem) that in some countries is forbidden for the use on veterinary medicine. The use of this class of antibiotics raise some ethical problem since it should be restricted to severe cases of human bacterial infection. This matter must be discussed further by the authors.

Authors’ response: Thank you for raising the ethical concern about using meropenem to treat melioidosis in animals. We agree with the reviewer that this concern needs to be discussed in the manuscript. We believe that this is a good opportunity for us to think about developing a guideline for treating melioidosis in animals. We have discussed this concern in the revised manuscript (Line 158-161).

Specific comments:

Title: I suggest to change as “Successful treatments and management of a case of canine melioidosis”

Authors’ response: We agree to change the title to “Successful treatments and management of a case of canine melioidosis” per the reviewer’s suggestion.

Line 49: please specify how the detection of E. canis and Anaplasma was done (Serology? PCR? Microscopic detection? If so please provides images). Moreover, I am not aware of Anaplasma canis species, rather Anaplasma phagocytophilum?

Authors’ response: Thank you for the comments. We used SNAP 4Dx Plus Test, a serology test kit, to detect ehrlichiosis and anaplasmosis in the dog. There was our mistake in typing a wrong species name. The correct specie is Anaplasma platys, not Anaplasma canis. Unfortunately, the hospital did not routinely take pictures made from this test.   

Line 94-95: again, specify how these agents were detected. Moreover, these two agents are not parasites but bacteria, please correct in the text.

Authors’ response: The statement has been corrected.

Line 142-142: this sentence claims a general conclusion which is however based on a single case report. I suggest to be more cautious.

Authors’ response: We agree with the reviewer’s concern. We have added a statement in the revised manuscript that “our report may be used as a guide for canine melioidosis case management with cautions that this report was based on a single case study”.

Reviewer 2 Report

In the manuscript by Khrongsee et al., the authors describe an unusual presentation of canine melioidosis and its successful treatment. The manuscript nicely describes the clinical presentation and treatment regimen that was employed here. Canine melioidosis is rarely reported, therefore, this manuscript provides a nice example, especially for treatment. I have no major issues with manuscript.

Minor corrections:

Line 100. “The dog was still remained” should be “The dog remained” Line 178. “appreciated the supports” should be “appreciate the support”

Author Response

Reviewer 2:

Comments and Suggestions for Authors

In the manuscript by Khrongsee et al., the authors describe an unusual presentation of canine melioidosis and its successful treatment. The manuscript nicely describes the clinical presentation and treatment regimen that was employed here. Canine melioidosis is rarely reported, therefore, this manuscript provides a nice example, especially for treatment. I have no major issues with manuscript.

Minor corrections:

Line 100. “The dog was still remained” should be “The dog remained” Line 178. “appreciated the supports” should be “appreciate the support”

Authors’ response: Thank you for checking the grammatical errors. Both statements have been corrected.

Reviewer 3 Report

This paper describes successful treatment of a dog with bacteraemicmelioidosis using a similar regimen to that used in humans.  It is not particularly surprising that the dog responded, although reports of successful treatment of animals with melioidosis are rare.

There is a need for a few corrections to the English, and in addition I note the following.

1.  I don't feel it is appropriate for a "guideline" to be based on a single case so suggest an alternative word is used in the abstract.

2.  Line 88.  What criteria (e.g. CLSI) were used to categorise the isolate as "susceptible" (line 88).

3.  Line 121.  The lack of reports of melioidosis in dogs probably reflects a lack of testing and surveillance so I would say "cases" rather than "incidence"  and briefly mention the lack of surveillance in animals.

4.  Line 150.  Meropenem is not given subcutaneously in humans so it should be made clear that this comment relates to animals.

Author Response

Reviewer 3:

Comments and Suggestions for Authors

This paper describes successful treatment of a dog with bacteraemic melioidosis using a similar regimen to that used in humans.  It is not particularly surprising that the dog responded, although reports of successful treatment of animals with melioidosis are rare.

There is a need for a few corrections to the English, and in addition I note the following.

I don't feel it is appropriate for a "guideline" to be based on a single case so suggest an alternative word is used in the abstract.

Authors’ response: We have adjusted the wording to not using the word “guideline” in the abstract. It now reads “ This communication presents a successful story of an attempt to treat and manage a case of canine melioidosis, a severe tropical disease caused by Burkholderia pseudomallei.”

Line 88.  What criteria (e.g. CLSI) were used to categorise the isolate as "susceptible" (line 88).

Authors’ response: We used NCCLS guidelines to categorize the isolate as “susceptible”. This has been mentioned in the text of the revised manuscript.

Line 121.  The lack of reports of melioidosis in dogs probably reflects a lack of testing and surveillance so I would say "cases" rather than "incidence"  and briefly mention the lack of surveillance in animals.

Authors’ response: We have adjusted the wording by using “cases” instead of “incidence” and provided a brief statement per the reviewer’s suggestions.

Line 150.  Meropenem is not given subcutaneously in humans so it should be made clear that this comment relates to animals.

Authors’ response: We have mentioned in the text that both meropenem and ceftazidime are not given subcutaneously in humans.